

# Aromatisation of steroids in the bivalve *Mytilus trossulus*

Anna Hallmann[1], Lucyna Konieczna[2], Justyna Swiezak[3],
Ryszard Milczarek[1] and Katarzyna Smolarz[3]

[1] Department of Pharmaceutical Biochemistry, Medical University of Gdańsk, Gdańsk, Poland
[2] Department of Pharmaceutical Chemistry, Medical University of Gdańsk, Gdańsk, Poland
[3] Department of Marine Ecosystem Functioning, University of Gdańsk, Gdynia, Poland

## ABSTRACT

In this study, we demonstrated the presence of the enzymatic complex able to perform aromatization (estrogen synthesis) in both, the microsomal and mitochondrial fractions of gills and gonads from *Mytilus trossulus*. Based on in vitro experiments, we highlighted the importance of temperature as the limiting factor of aromatisation efficiency (AE) in mussels. After testing range of temperatures (4–23 °C), the highest AE was found during incubation at 8 °C and pH 7.6 (41.66 pmol/h/mg protein in gills and 58.37 pmol/h/mg protein in gonads). The results were confirmed during field studies where the most efficient aromatisation occurred in bivalves collected in spring while the least effective in those collected in winter. During in vitro studies, AE turned out to be more intensive in female gonads than in male gonads. The process was also more intensive in mitochondrial fraction than in microsomal one (62.97 pmol/h/mg protein in male gills and 73.94 pmol/h/mg protein in female gonads). Enzymatic complex (aromatase-like enzyme) catalysing aromatisation in mussels was found to be insensitive to inhibitory effect of selective inhibitors of mammalian aromatase such as letrozole and anastrazole, suggesting its different structure from vertebrate aromatase. Further in vivo studies using $^{13}$C-labeled steroids at 8 °C temperature window confirmed that bivalves are able to uptake testosterone and androstenedione from the ambient environment and metabolise them to estrone and 17β-estradiol thus confirming endogenous estrogen' synthesis.

## INTRODUCTION

Since the presence of various sex steroids has long been confirmed in many species of marine bivalves, the two main questions in bivalves endocrinology are related to (i) the role of steroids and (ii) their origin in these organisms. Indeed, some studies provide evidence that sex steroids serve various physiological functions in bivalve tissues. For example, fluctuations of 17β-estradiol (E2) and testosterone (T) levels in *Sinonovacula constricta* were related to the reproductive cycle of the clams indicating their possible role as endogenous modulators of the gametogenetic cycle (*Yan et al., 2011*). In a study of *Mezghani-Chaari et al. (2017)* an exposure of *Ruditapes decussatus* to high 17β-estradiol

Corresponding author
Katarzyna Smolarz,
oceksm@univ.gda.pl

doses resulted in sex-specific disorders in gametogenesis. In *Li et al. (1998) Crassostrea gigas* was exposed to E2 which promoted vitellin formation in the ovarian tissue. In *Mytilus edulis* an exposure to 17β-estradiol induced various alterations in the mRNA expression of monoamine serotonin receptor and cyclooxygenase that correlated with the bivalves reproductive stage (*Cubero-Leon et al., 2010*). Yet, comparable exposure of *M. edulis* to E2 led to increased tissue concentrations of free and total estradiol, but induction of VTG or estrogen receptor (ER) gene expression in gonads was not confirmed (*Puinean et al., 2006*). Similarly, in other studies no or very little endocrine-related reproductive effect was identified (*Ketata et al., 2008*; *Markov et al., 2009*; *Morthorst et al., 2014*). Hence, *Scott (2013)* in a critical review concluded that there is no indisputable bioassay evidence that vertebrate sex steroids have endocrinological or reproductive roles in molluscs. In spite of that, an association between gametogenesis stage and steroid content was reported with maximal steroids concentration often observed during reproduction peak and minimal concentration during low stage of gametogenesis (*Ni, Zeng & Ke, 2013*; *Wang & Croll, 2007*; *Liu, Li & Kong, 2008*; *Ciocan et al., 2010*; *Morthorst et al., 2014*; *Hallmann et al., 2016*). In *Mytilus trossulus* from the Gulf of Gdańsk (Baltic Sea, Poland), seasonal differences in E2 and T concentrations in gonads and gills were also found, but no direct relation between steroids level and stage of gonad development was confirmed (*Zabrzańska et al., 2015*; *Hallmann et al., 2016*; *Smolarz et al., 2018*). Instead, a strong correlation between water temperature and steroids level was observed (*Smolarz et al., 2018*) suggesting temperature as an important factor influencing tissue steroids content.

With regards to steroids such as estrogens, neither their origin nor their role in bivalves has been confirmed because steroidogenesis is not yet indubitably documented in this group of organisms. Known similarities in functioning of endocrine system in vertebrates and invertebrates include various signals of transduction to appropriate target sites for regulating gene expression (*Janer et al., 2005*). In higher organisms, the cytochrome P450 aromatase (CYP19)—a member of a large superfamily of cytochrome P450 enzymes—is involved in steroid biosynthesis and is expressed in estrogen-producing cells (*Hall, 1985*). Aromatase is responsible for the conversion of the androgens; androstenedione and testosterone into the estrogens: estrone and 17β-estradiol, respectively (*Thompson & Siiteri, 1974*; *Kellis & Vickery, 1987*; *Yoshida & Osawa, 1991*). Aromatization of androgens to estrogen occurs after multiple oxygenation processes, loss of the methyl group at C-19 and the elimination of the 1β hydrogen (*Numazawa, Yoshimura & Oshibe, 1998*). During aromatization, three moles of oxygen and three moles of NADPH for every mole of metabolized steroid substrate are used, resulting in three $H_2O$ molecules (*Osawa et al., 1987*). In vertebrates, aromatase is mainly present in the microsomal fraction (*Simpson et al., 2002*). However, earlier studies report that this enzyme can also be found in the mitochondria of human placenta (*Finkelstein, Neiman & Mizrahi, 1985*; *Meigs & Moorthy, 1984*). Aromatase activity level in fish equals to 16 pmol/h/mg protein (*Orlando, Davis & Guillette, 2002*), in birds equals to 50 pmol/h/mg protein (*Foidart et al., 1998*) and in human placenta 95 pmol/min/mg protein (*Milczarek et al., 2008*).

Despite determination of androgens and estrogens in bivalve tissues, the occurrence of aromatization in this group of organisms is still unclear since most studies suggest

that aromatase activity is either absent or at a very low level (*Scott, 2012*). In marine invertebrates, determination of aromatase activity is based on methods such as ELISA test, in which the content of final product is measured (*Tinwell et al., 2011*). This approach, however, lacks specificity since the reaction can be catalysed by aromatase as well as by other unidentified enzymes belonging to cytochrome P450 superfamily. Hence, the usage of antibodies against mammalian CYP19 in immunohistological staining in scattered single bivalve cells may spot non-specific cross-reactivity. Commercially available kits such as aromatase (CYP19A) Activity Assay Kit (Fluorometric, BioVision) seem to be suitable when applied to human tissue only, as they contain specific inhibitors of human aromatase that may not inhibit the reaction in bivalves. However, a method allowing to measure aromatization efficiency (AE) was described by *Lavado, Janer & Porte (2006)*. It uses isotope-labeled substrate utilized by the $1\beta$-$^3$H androstenedione. In the presence of aromatase, a quantified amount of $^3$H$_2$O is produced from isotope-labeled substrate allowing for quantification of aromatization. Also here, non-specific cross-reactivity cannot be excluded since the reaction can be catalysed by aromatase as well as by a random enzyme belonging to the group of the cytochrome P450 family that contributes to oxygenations and loss of the methyl group at C-19 position. Aromatase (or aromatase-like) activity measured using isotopic method in *C. gigas* was as low as six fmol/h/gram of wet weight (*Le Curieux-Belfond et al., 2001*) and therefore close to the measurability threshold. AE measured in gonads and digestive gland of *M. edulis* was also low (one and three pmol/h/mg of protein at maximum, respectively) (*Lavado, Janer & Porte, 2006*). In addition, aromatase (CYP19) gene orthologue first appeared in a direct ancestor of the chordates—amphibians and there is no information about CYP19 gene in invertebrates available (*Callard et al., 2011*). Hence, *Scott (2012)* in a critical appraisal gave a strong reasoning against the hypothesis of endogenous steroid origin in molluscs. Nevertheless, comparative phylogenetic analyses suggest the presence of aromatase gene also in lower organisms (*Castro, Santos & Reis-Henriques, 2005*).

Since vertebrate aromatase is very sensitive to the inhibitory effects of certain drugs such as letrozole, anastrozole, and ketoconazole (KZ), one way of comparing the similarity of an enzyme catalysing aromatization in various species to vertebrate aromatase is by using targeting selectivity of known aromatase inhibitors. Letrozole inhibits aromatase activity in a variety of mammalian tissues by iron binding in hem moiety of CYP-450 which is a subunit of the aromatase enzyme complex (*Haynes et al., 2003*; *Bhatnagar, 2007*). Anastrozole, by competitive inhibition achieved by direct and reversible bond to aromatase, blocks the conversion of androgens to estrogens (*Hortobagyi & Buzdar, 1998*). KZ, characterized by the widest work spectrum, is commonly used as a synthetic antifungal pharmaceutical. It inhibits in a dose-dependent matter ovarian $3\beta$-hydroxysteroid dehydrogenase and 17-hydroxylase, essential enzymes for the formation of C-19 steroids (*Kunio et al., 1986*). It is also used as an irreversible steroidal aromatase inhibitor in pharmacotherapy. KZ also inhibits cholesterol side chain cleavage enzymes in the adrenal and testis, thus affecting androgen biosynthesis (*Kunio et al., 1986*; *DiMattina et al., 1988*).

The *M. edulis* complex belongs to strictly gonochorous species reaching sexual maturity at the age of 1 year with typical sex ratio around one (*Kautsky, 1982*; *Newell et al., 1982*).

The reproduction of the blue mussel from the Baltic Sea, like many other temperate species, follows a seasonal pattern since it is correlated with temperature and food availability (*Kautsky, 1982*; *Kautsky & Evans, 1987*). Early phase of gametogenesis occurs in late winter and/or early spring and the main spawning event takes place between May and July, but a second spawning peak can also be observed in autumn. To our knowledge, there is no data confirming the presence of hormone receptors initiating multiple signalling pathways and ultimately leading to sex determination and/or gametogenesis in bivalves. There is, however, a confirmed relationship between ambient water temperature and steroid content in *M. trossulus* (*Smolarz et al., 2018*). Additionally, the blue mussels are characterized by double uniparental inheritance (DUI) in which females inherit maternal mitochondria only while males inherit maternal and paternal ones. Since in pair mating this mtDNA biparental inheritance was associated with strong sex ratio bias, a relationship between DUI and sex determination in genus *Mytilus* was proposed by *Passamonti & Ghiselli (2009)* and *Zouros et al. (1994)*. Still little is known about interactions between reproduction pattern, sex determination and steroids in bivalves, hence a regulative role of estrogens in DUI-related sex determination cannot be excluded.

The main purpose of this study was therefore to confirm the occurrence of endogenous synthesis of steroids in marine bivalves by addressing aromatization in in vitro and in vivo studies using *M. trossulus* as a model species. In particular, we aimed at analysing (i) the effect of temperature (season) and pH on the effectiveness of aromatization; (ii) AE in microsomal and mitochondrial fractions of gills and gonads; (iii) similarity of an enzyme catalysing aromatization in bivalves to mammalian aromatase using targeting selectivity of known aromatase inhibitors and (iv) sex- and tissue-related differences in aromatization efficiency.

## MATERIAL AND METHODS

### Chemicals and reagents

Dexamethasone (internal standard), 4-Androstene-3,17-dione-2,3,4-$^{13}C_3$, Estrone-2,3,4-$^{13}C_3$, 17β-Estradiol-2,3,4-$^{13}C_3$, Testosterone-2,3,4-$^{13}C_3$, Anastrozole, Letrozole, KZ, protease from *Bacillus licheniformis* (subtilisin), formic acid, acetone, methanol, acetonitrile, hexane, methylene chloride, charcoal activated, Tris, HCl, $KH_2PO_4$, $K_2HPO_4$, KCl, NADPH, glycerol, BSA, scintillation cocktail, and Supel$^{TM}$-Select SPE HLB (six mL, 200 mg) columns were provided by Sigma-Aldrich (St. Louis, MO, USA). 1β-$^3$H Androstene-3,17-dione (30 Ci/mmol) was obtained from Perkin-Elmer Life Science (Boston, MA, USA).

### Sampling and tissue preparation

The blue mussels *M. trossulus* were collected by dredging from a sampling station located on the coastline of the Gulf of Gdańsk, Poland (N: 54°40′00″E: 18°33′55″) at 10.0 m depth. The shell length of collected individuals varied from 25 to 30 mm which corresponded to an age range of 3–4 years and indicated their sexual maturity. Next, mussels were transported to the laboratory and kept in aerated aquaria in conditions resembling natural when sampled (temperature and salinity).

In vitro analyses were performed on organisms collected in the period from 2012 to 2018. After dissection, gills and gonads were separated and mitochondria and microsomes were isolated, frozen and kept in −80 °C until the analysis. The analyses of temperature- and pH-related changes in AE in bivalve tissues were performed between 2012 and 2014 on 150 individuals. Seasonal analyses of AE were performed on approximately 400 blue mussels sampled in May 2012, July 2012, November 2012, and February 2013 (~100 mussels analysed per season). In March 2015, the effect of aromatase inhibitors was studied on 100 organisms. Sex- and tissue-related differences in AE were performed on 100 organisms sampled in April 2016. Sexing was based on a small subsample of gonadal tissue placed on the microscope slide with saline solution and covered by cover slide. Prepared smears were analysed under light microscopy for the presence of ovaries (oocytes) or testis (spermatocytes) and classified accordingly as females or males. In vivo analyses were performed on individuals collected in April 2018. Exposure to $^{13}C_3$–labeled steroid mixture of blue mussels was performed without (in the first experiment) and with (in the second experiment) sexing after which bivalves were dissected and the whole soft tissue was frozen in liquid nitrogen and stored at −80 °C.

Temperature, pH, seasonal, sex, and tissue-related aromatisation efficiency (AE) based on in vitro experiments

### Preparation of gills and gonadal mitochondrial and microsomal fraction

Gonadal and gill tissues were pooled separately in order to obtain 25 mL of each tissue volume. The tissue was manually homogenized in a glass Potter–Elvehjem homogenizer with Teflon pestle in ice-cold 100 mM $KH_2PO_4/K_2HPO_4$ buffer (pH 7.4) containing 0.15 M KCl. Homogenates were centrifuged at $500{\times}g$ for 15 min for removing nuclei and cell membranes. Next, centrifugation at $15,000{\times}g$ for 30 min was performed to obtain mitochondria. Obtained supernatant was further centrifuged at $100,000{\times}g$ for 1 h for gaining microsomal pellet. The mitochondrial and microsomal pellets were diluted in 100 mM $KH_2PO_4/K_2HPO_4$ buffer (pH 7.4) containing 0.15 M KCl and 20% (w/v) glycerol. Suspended mitochondrial fraction (volume ranging from two to three mL) containing 20–50 mg of protein per one mL was divided into 100 μL subsamples. Similarly, microsomal fraction (volume ranging from one to two mL) containing ~20 mg of protein per one mL was divided into 100 μL subsamples. The subsamples (containing at least one mg of protein) were then frozen in −80 °C for further analyses. Protein content was determined after *Lowry et al. (1951)* with bovine serum albumin as a standard. Temporal differences in AE were analysed in both tissue types based on microsomal isolates pooled from various individuals.

### Determination of aromatization in model species

Aromatisation efficiency was determined in the mitochondrial and microsomal fractions of both, gill, and gonadal tissues, by the tritiated water release method as described in *Shimizu et al. (1995)* with some modifications. Both fractions (one mg) were incubated in glass tubes in temperature gradient 4–23 °C (tested temperatures were 4, 5, 6, 7, 8, 9, 10, 11, 18, 23 °C) for 1 h in a final one mL volume in the presence of 100 mM Tris–HCl

buffer of various pHs (pH 6.0, 6.6, 7.0, 7.6, 8.0, 8.6, and 9.0), 10 μM (1β-$^3$H)-androstenedione (with specific activity ranging from 150 to 200 DPM/pmol) and 200 μM NADPH. First, temperature-based changes in AE were studied in order to select temperature window in which the process is most efficient. Next, the effect of pH on AE was analysed. In order to study similarity of an enzyme catalysing aromatization in bivalves to vertebrate aromatase, one mg of microsomal fraction isolated from gonads characterized by high AE was incubated for 1 h in the presence of letrozole, anastrozole, and KZ at concentrations of 0.1, 0.25, 0.5, and 1 mM.

Next, organic metabolites and the excess of substrate were removed from the aqueous phase by extraction with methylene chloride (three mL). The samples were further centrifuged at 4,000×$g$ for 15 min. After centrifugation, 5% activated charcoal (two mL) was added to the aqueous phase, the solution was shaken for 2 min at room temperature and covered probes were left overnight. The next day, the solution was centrifuged at 4,000×$g$ for 1 h and afterward the collection of supernatant was performed. The supernatant was mixed with four mL of scintillation liquid and the solution was analysed using Beckman LS 6000 IC counter. AE was assessed as tritium release to water from 1β-$^3$H of androstenedione upon aromatization and was expressed as disintegrations per minute per hour per milligram protein and calculated as pmol/h/mg protein.

## Biosynthesis of estrogens—in vivo study

A total of 15 blue mussels collected in spring 2018 were kept at 6 °C and further individually transferred to glass beakers with aerated artificial seawater (200 mL) treated with 500 μL $^{13}$C$_3$–labeled steroid mixture (4-Androstene-3,17-dione-2,3,4-$^{13}$C$_3$ or Testosterone-2,3,4-$^{13}$C$_3$ in acetonitrile) at the final concentration of 250 ng/mL and 1 ml of algae suspension. The control sample consisted of specimens transferred to glass beakers containing 200 mL of seawater, 500 μL of acetonitrile, and 1 ml of algae suspension. Next, beakers were placed in a water bath at a temperature of 6 °C for 2 h. After the 2-h incubation period the temperature was gradually increased to 8 °C and the incubation continued for another 22 h. After incubation, mussels were dissected, the whole soft tissue was frozen in liquid nitrogen and stored at −80°C. Subsequently, the soft tissue was prepared for solid phase extraction followed by analysis of steroids content using LC-MS/MS method. The above-described exposure to labeled androgens was repeated in identical conditions using 10 additional mussels on which, after soft tissue dissection, sexing was performed.

### Extraction procedure

Sample extraction method was adopted from *Vanhaecke et al. (2011)* and further modified as described below. Enzymatic digestion with subtilisin was used in order to support tissue homogenization via protein cleavage (*Blasco, Van Poucke & Van Peteghem, 2007*). Frozen sample was transported into 20 mL glass flasks and thawed for 15 min in the presence of subtilisin enzyme (0.4 mg/1 g tissue). Afterward, samples were microwaved for 60 s at 100 W, the content was transferred to 50 mL polypropylene tubes and

homogenized using an Ultra-turrax instrument (Janke & Kunkel, IKA-Labortechnik, Staufen, Germany) for 1 min. After adding five mL of methanol, samples were vortexed for 1 min and centrifuged ($5{,}000 \times g$, 10 min, 4 °C). Methanolic supernatant was collected and distilled water (10 mL) was added to the extract. The extract was concentrated on a Supel™-Select SPE HLB (six mL, 200 mg) column. The cartridge was washed with water and hexane. The analytes were eluted with methanol. Upon evaporation under a stream of nitrogen, the dry residue of the eluate was dissolved in methanol/water mixture (65:35, v/v).

### Steroids-$^{13}C_3$ identification—chromatographic and MS conditions

Chromatographic separation was achieved using LC system coupled with MS (LC-MS/MS-8050 Shimadzu, Japan). The mobile phase was composed of 0.1% formic acid in ultrapure water (phase A) and acetonitrile with addition of 0.1% formic acid (phase B). Analytes of interest were separated using a C-18 core-shell type column (Poroshell 120 EC-C18; Agilent Technologies, St. Clara, CA, USA) with dimensions 3.0 × 100 mm, 2.7 μm particle size with flow rate of 0.5 mL/min. To optimize the gradient elution phase B was initially set at 31.1% level with successive increase to 55.1% during a 12 min period. Next, the proportion of eluent B was increased to 100% during 0.9 min and then remained isocratic for a period of 3 min. The autosampler' storage compartment was kept at 15 °C to increase sample stability. The injected sample' volume was one μL. To minimize carry-over effect, the needle was flushed with methanol after injection. The total time of analysis was 16 min. Next, the column rebalanced to the initial conditions. The injector needle was washed between injections with 50:50 acetonitrile-water and the needle seat back was flushed with acetonitrile-water mixture (50:50, v/v) at 35 μL/s for 2 s to evade contamination. LC-MS/MS was operated in the positive ion mode equipped with electrospray ionization source working with temperature-optimized conditions of the following components: interface 300 °C, heater block 200 °C, desolvation line (DL) 200 °C, and gas flows: drying gas ($N_2$) 10 L/min, nebulising gas ($N_2$) three L/min and heating gas (air): 10 L/min. Capillary voltage was maintained at three kV. Data were acquired in scheduled MRM mode using a detection window. The device was calibrated and tuned in agreement with procedures recommended by the manufacturer. Data acquisition and analysis of chromatograms were processed with LabSolution software (Shimadzu, Japan). The acquisition range in MS scanning mode ranged from 50 to 400 m/z with a spectra rate of one Hz. Each sample was analysed in triplicate. Details concerning *m/z* values of precursor and product ions and exemplary chromatograms for all analysed steroids are presented in Table 1 and Fig. 1. The LOQ of the LC-MS/MS method was determined at 0.08 ng/g for E1-$^{13}C_3$ and 17βE2-$^{13}C_3$, 0.06 ng/g for A-$^{13}C_3$ and T-$^{13}C_3$.

## Statistical analysis

The statistical significance of differences between groups was verified with the non-parametric Kruskal–Wallis ANOVA test and the significance level was set at $p < 0.05$. All statistical analyses were carried out in STATISTICA 13.0 software.

Table 1 Isotopically $^{13}C_3$-labeled standards of steroids.

| Steroid standards | Molecular weight | Retention time | Precursor ion m/z | Product ion m/z | Collision energy (CE) [eV] |
|---|---|---|---|---|---|
| Dexamethasone (internal standard) | 392.46 | 3.51 | 393.10 | 237.15 | −19 |
| | | | | 355.05 | −12 |
| | | | | 373.20 | −10 |
| 4-Androstene-3, 17-dione-2,3,4-$^{13}C_3$ | 289.39 | 7.47 | 290.10 | 100.15 | −22 |
| | | | | 272.15 | −16 |
| | | | | 112.15 | −25 |
| Testosterone-2,3,4-$^{13}C_3$ | 291.40 | 5.89 | 292.10 | 100.15 | −23 |
| | | | | 112.05 | −25 |
| | | | | 256.30 | −17 |
| Estrone-2,3,4-$^{13}C_3$ | 273.34 | 7.25 | 274.00 | 256.10 | −14 |
| | | | | 160.05 | −18 |
| | | | | 136.10 | −22 |
| 17β-estradiol-2,3,4-$^{13}C_3$ | 275.36 | 5.57 | 258.00 | 163.10 | −19 |
| | | | | 58.15 | −24 |
| | | | | 162.10 | −18 |

**Note:**
Dexamethasone (internal standard), 4-Androstene-3,17-dione-2,3,4-13C3, Testosterone-2,3,4-$^{13}C_3$, Estrone-2,3,4-$^{13}C_3$, 17β-estradiol-2,3,4-$^{13}C_3$. Detailed data concerning *m/z* values of precursor and product ions for all analyzed steroids.

## RESULTS

### The effect of temperature and pH on AE

Differences in aromatase efficiency were found when the process was analysed in various temperature conditions (Dataset S1). In the temperature range from 4 to 5 °C AE was low both in gills (10% of maximal activity recorded) and in gonads (5% of maximal activity) of the blue mussel. An increase of aromatase efficiency (pmol/h/mg protein) was observed in temperatures above 5 °C. In the temperature range from 8 to 9 °C AE was the highest with 41.66 pmol/h/mg protein in the microsomal fraction of gills and 58.37 pmol/h/mg protein in the microsomal fraction of gonads (Fig. 2A). Aromatization in temperatures above 8 °C resulted in a lower substrate consumption and thus decreased production of tritiated water (lower efficiency). Room temperature (23 °C) resulted in negligible amounts of $^3H_2O$ produced, that is, 8% (gills) and 12% (gonads) of maximal AE (Fig. 2A). Since the highest AE occurred in 8–9 °C, pH related differences in AE were performed in controlled temperature conditions using gill and gonad subcellular fractions (Dataset S2). The process was stable in pH range from 7.6 to 9.0 for gills and showed similar levels of the AE—41.83 pmol/h/mg protein on average. Similarly, in microsomes from gonads aromatization level was stable and led to the production of about 51.89 pmol/h/mg protein in the pH range from 7.0 to 8.6 (Fig. 2B). In pH levels from 6.0 to 7.0 AE was lower than in the remaining pHs (Fig. 2B). No statistically significant differences in AE measured in pH levels from 7.6 to 9.0 were found, thus in further analyses physiological pH 7.6 was used. Each subsequent AE measurements in the microsomes and mitochondria isolated from the bivalve tissues were conducted in optimal conditions, at 8 °C and pH 7.6. AE was also checked in conditions such as the absence of NADPH and in denaturated microsomes. In both cases, the efficiency of AE decreased by min. 70% (Table S1).

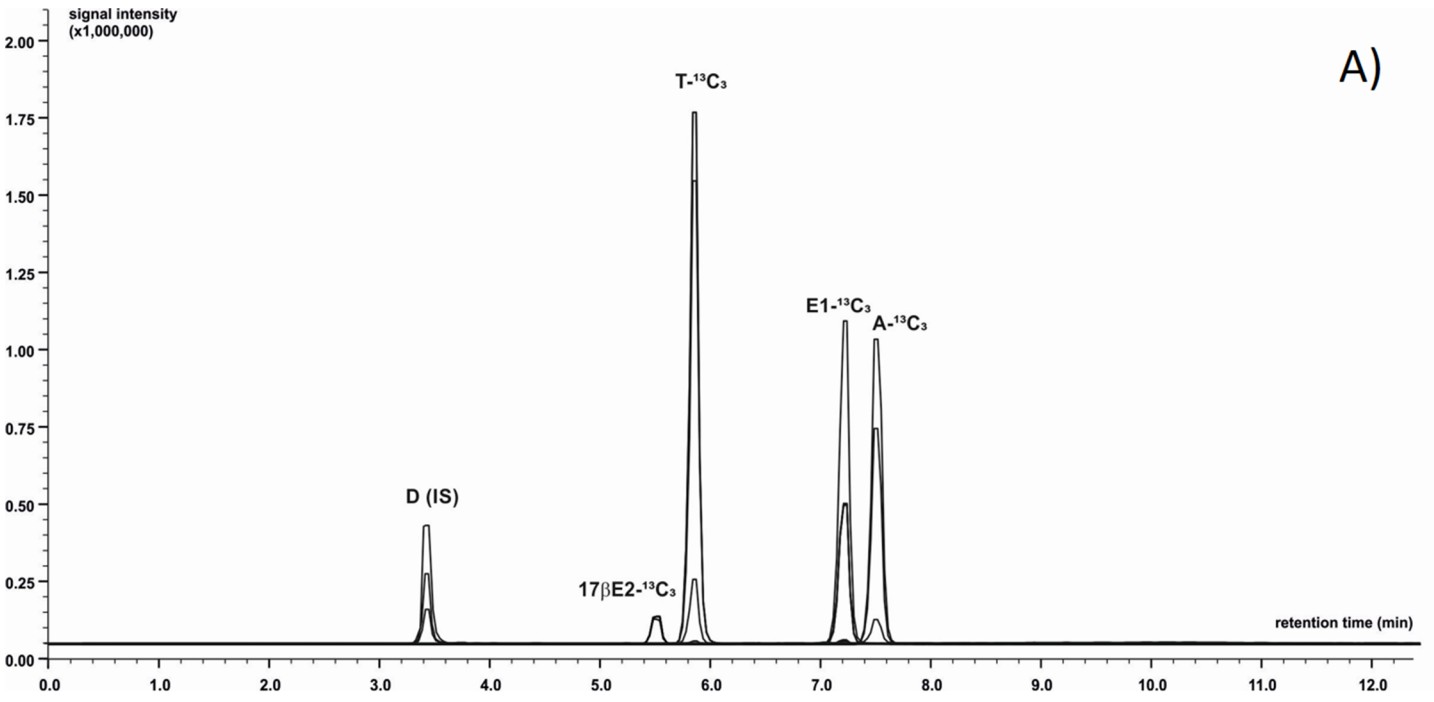

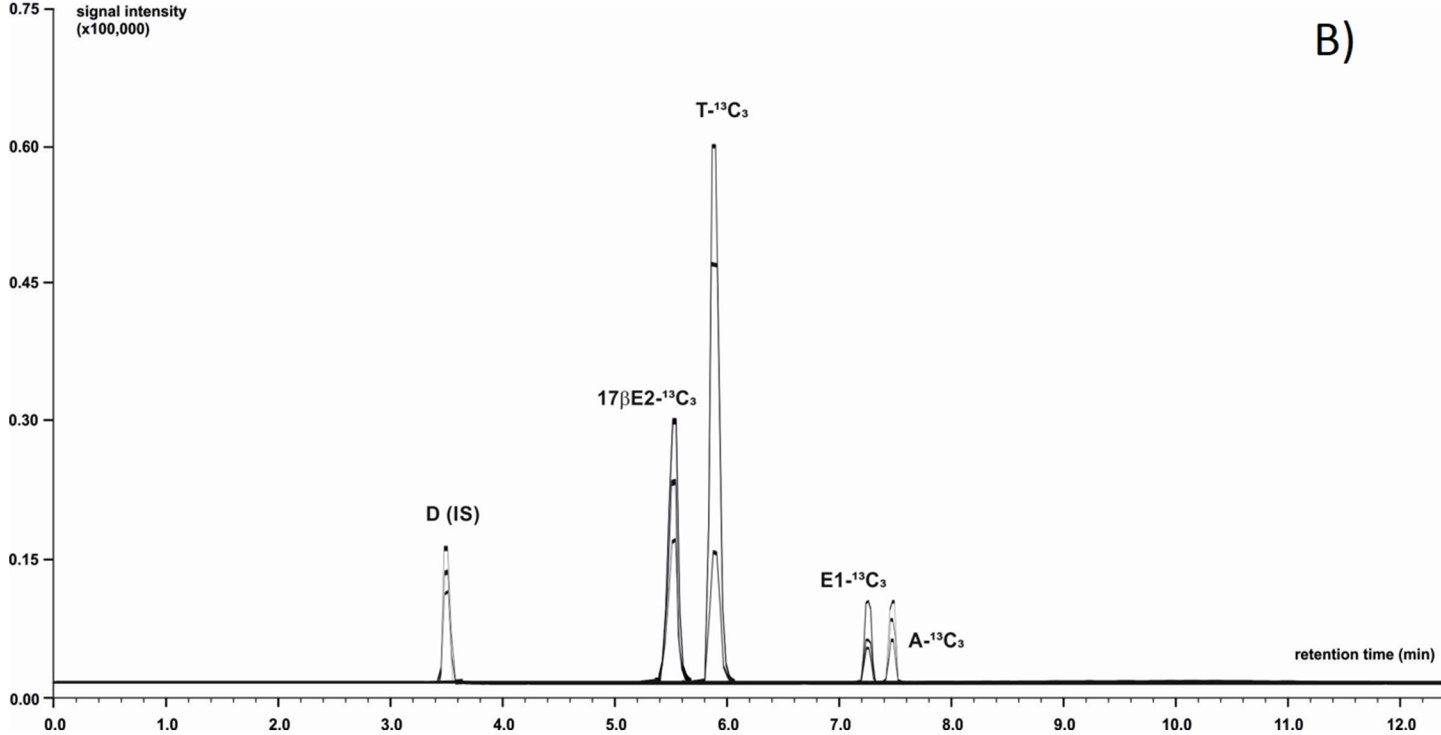

**Figure 1 Identification of $^{13}C_3$-labeled steroids by LC-MS/MS.** Positive ion chromatogram of steroid standards (A) was compared to chromatogram obtained from extract of *M. trossulus* exposed to Testosterone-2,3,4-$^{13}C_3$ over 24 h (B). Abbreviations: D, Dexamethasone (internal standard); A-$^{13}C_3$, 4-Androstene-3,17-dione-2,3,4-$^{13}C_3$; T-$^{13}C_3$, Testosterone-2,3,4-$^{13}C_3$; E1-$^{13}C_3$, Estrone-2,3,4-$^{13}C_3$; 17βE2-$^{13}C_3$, 17β-estradiol-2,3,4-$^{13}C_3$.

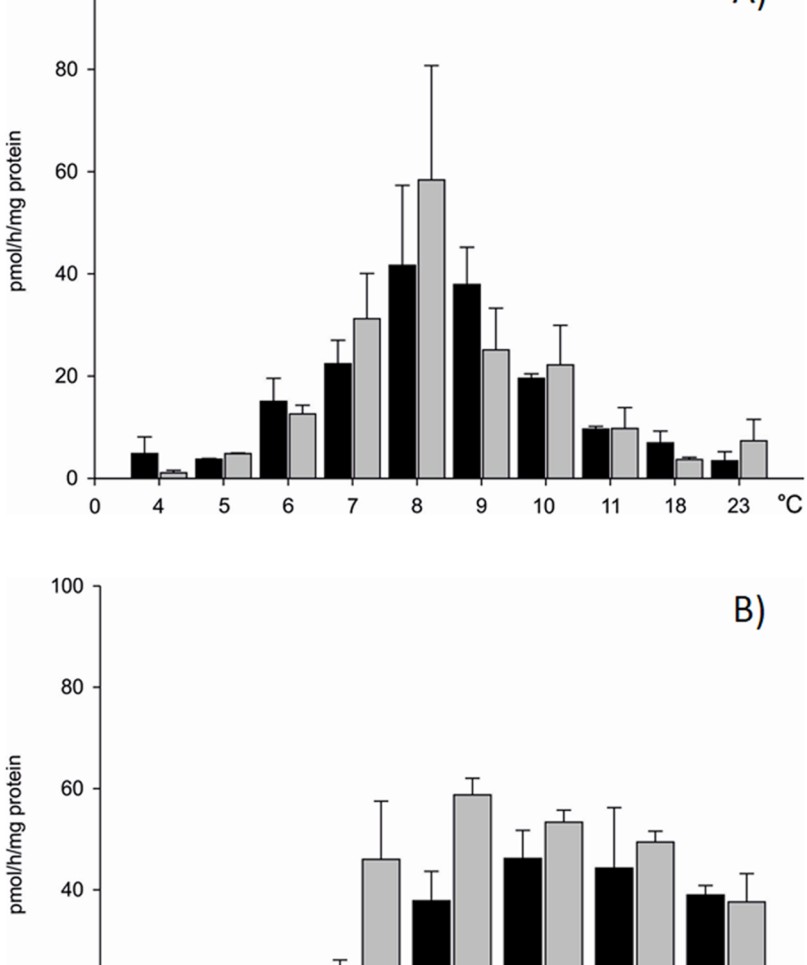

**Figure 2 Aromatization efficiency in microsomal fraction isolated from gills (black column) and gonads (gray column) of *M. trossulus* (A) temperature dependent efficiency, (B) pH dependent efficiency.** Data presented as mean ± SD ($n = 5$). 

## Seasonal changes in AE

Statistically significant seasonal differences in AE were found (Fig. 3). The most efficient aromatization was recorded in mussels collected in spring in microsomal fraction originating from both analysed tissues; 31.75 ± 7.51 pmol/h/mg protein in gill and 39.34 ± 4.25 pmol/h/mg protein in gonad. In mussels collected in summer and autumn seasons AE decreased by 50% when compared to spring and equalled to 13.84 ± 3.27 pmol/h/mg protein in gills and 15.29 ± 2.28 pmol/h/mg protein in gonads. The lowest AE equaled to 3.17 ± 1.22 pmol/h/mg protein in gills and 1.84 ± 1.02 pmol/h/mg protein in gonads and was measured in winter season (Fig. 3; Dataset S3). No statistically significant

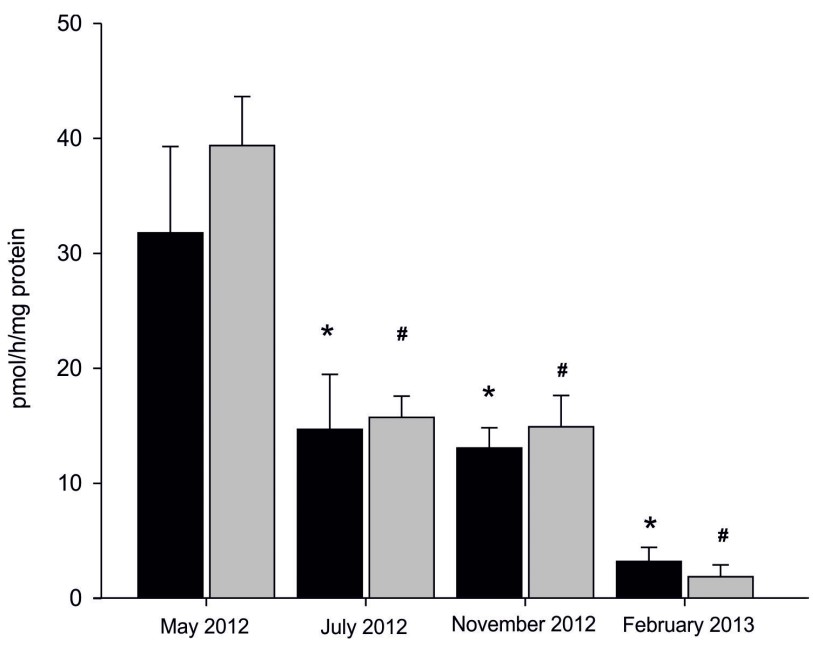

**Figure 3 Seasonal differences in AE in microsomes isolated from gills (black column) and gonads (gray column) of *M. trossulus*.** Data presented as mean ± SD ($n = 3$). Significant differences with respect to the "May 2012" groups indicated by *,#$p < 0.05$ (Kruskal–Wallis ANOVA).

differences in AE between analysed tissues in each season were found except for winter, where AE, despite being at the lowest level, was twice as high in gills as in gonads. However, a trend toward more efficient aromatization in blue mussels gonads compared to gills can be seen in all seasons but winter.

## Sex-related differences in AE in mitochondrial and microsomal fractions from gills and gonads

Sex- and fraction-related differences in AE were also confirmed. In females, AE was higher in the mitochondrial fraction compared to the microsomal fraction isolated from both tissues. Activity of an aromatase-like enzyme complex catalysing aromatization in gills was 33.73 ± 6.59 pmol/h/mg of protein in microsomes and 43.12 ± 2.70 pmol/h/mg of protein in the mitochondrial fraction. In gonadal tissue, its activity equalled to 44.78 ± 1.90 pmol/h/mg of protein in microsomes, while in the mitochondria 1.6-fold higher activity was measured (73.94 ± 2.41 pmol/h/mg protein). Similarly, in males a higher AE in the mitochondrial fraction compared to the microsomal one was found. In gills, aromatization occurring at 29.77 ± 8.41 pmol/h/mg of protein was confirmed in microsomes, and at 62.97 ± 6.56 pmol/h/mg of protein in the mitochondria. In gonads, higher AE was found in the mitochondrial fraction (56.71 ± 1.85 pmol/h/mg protein) compared to the microsomes (33.91 ± 6.60 pmol/h/mg protein). In females, statistically significant differences in AE between tissues were found. In microsomal fraction of gonads AE was 30% higher than in the same fraction isolated from gills. Similarly, in mitochondrial fraction isolated from gonads AE was 1.7 times higher than in mitochondrial fractions

from gills. In males, no statistically significant differences in AE measured in both tissues were found in microsomal fraction. However, AE was higher in gill mitochondria than in gonadal mitochondria ($p < 0.05$). Sex-related differences in AE included 1.3 times higher AE in female gonads when compared to male gonads (both fractions) and 1.5 times higher aromatization in mitochondrial fraction isolated from male gills when compared to female gills (Fig. 4; Dataset S4).

## The effect of inhibitors on AE in gonads of *M. trossulus*

The effect of inhibitors on AE is presented in Dataset S5 (raw data). In the presence of one mM letrozole, a 100% increase in AE in the subcellular fraction of gonads was found even in the highest concentration of the inhibitor ($81.57 \pm 3.62$ pmol/h/mg of protein). Also in the presence of 0.5 mM anastrozole, there was an evident and significant increase in AE (almost 40%, $66.43 \pm 1.83$ pmol/h/mg of protein) when compared to the control ($p < 0.05$). The inhibiting effect of anastrozole was only observed at the concentration of one mM, resulting in inhibition of aromatization by 50% ($21.11 \pm 0.62$ pmol/h/mg of protein) when compared to the control (Fig. 5). The inhibitory effect of KZ was confirmed in all tested concentrations. Inhibition was linear and the efficiency of the process gradually decreased along used concentration gradient (Fig. 5).

## Estrogens synthesis in vivo and identification of other steroids via mass spectrometry

In 90% of individuals (18 out of 20) both androgens-$^{13}C_3$ and estrogens-$^{13}C_3$ were identified. In bivalves exposed to Testosterone-2,3,4-$^{13}C_3$ (T-$^{13}C_3$) both T-$^{13}C_3$ and 4-Androstene-3,17-dione-2,3,4-$^{13}C_3$ (A-$^{13}C_3$) were identified but at different concentrations. High concentration of T-$^{13}C_3$ ($8.48 \pm 3.29$ ng/g wet weight) and eight times lower concentrations of 4-Androstene-3,17-dione-2,3,4-$^{13}C_3$ ($0.96 \pm 0.30$ ng/g w. w.) were detected (Fig. 6; Table S2). Similarly, an exposure to A-$^{13}C_3$ resulted in detection of both, T-$^{13}C_3$ and A-$^{13}C_3$ in analysed tissues. In mussels exposed to T-$^{13}C_3$ the presence of 17β-estradiol-2,3,4-$^{13}C_3$ ($0.97 \pm 0.44$ ng/g w. w.) was confirmed, whereas in those exposed to A-$^{13}C_3$, Estrone-2,3,4-$^{13}C_3$ ($2.75 \pm 1.72$ ng/g w. w.) was found (Fig. 6; Table S2). The analysis of remaining labeled steroids revealed mutual and reversible conversion of androgens with A-$^{13}C_3$ transformed into T-$^{13}C_3$. Similarly, reversible transformation of estrogens was observed with E1-$^{13}C_3$ converted to 17βE2-$^{13}C_3$. No sex-related difference in the steroid uptake or synthesis was found (Tables S2A and S2B).

## DISCUSSION

In marine invertebrates, activities of selected steroidogenic enzymes (3β-hydroxysteroid dehydrogenase, 17α-hydroxylase, aromatase, and aromatase-like enzyme) measured in microsomal fraction from various tissues were detected at very low levels, or not detected at all. For example, in coral *Pocillopora damicornis* aromatase-like activity was determined by measuring the conversion of testosterone to 17β-estradiol using ELISA test and the resulting activity was calculated as 10–1,000 fg min/mg protein (*Rougée, Richmond & Collier, 2015*). In sea urchin *Paracentrotus lividus* P450-aromatase activity measured

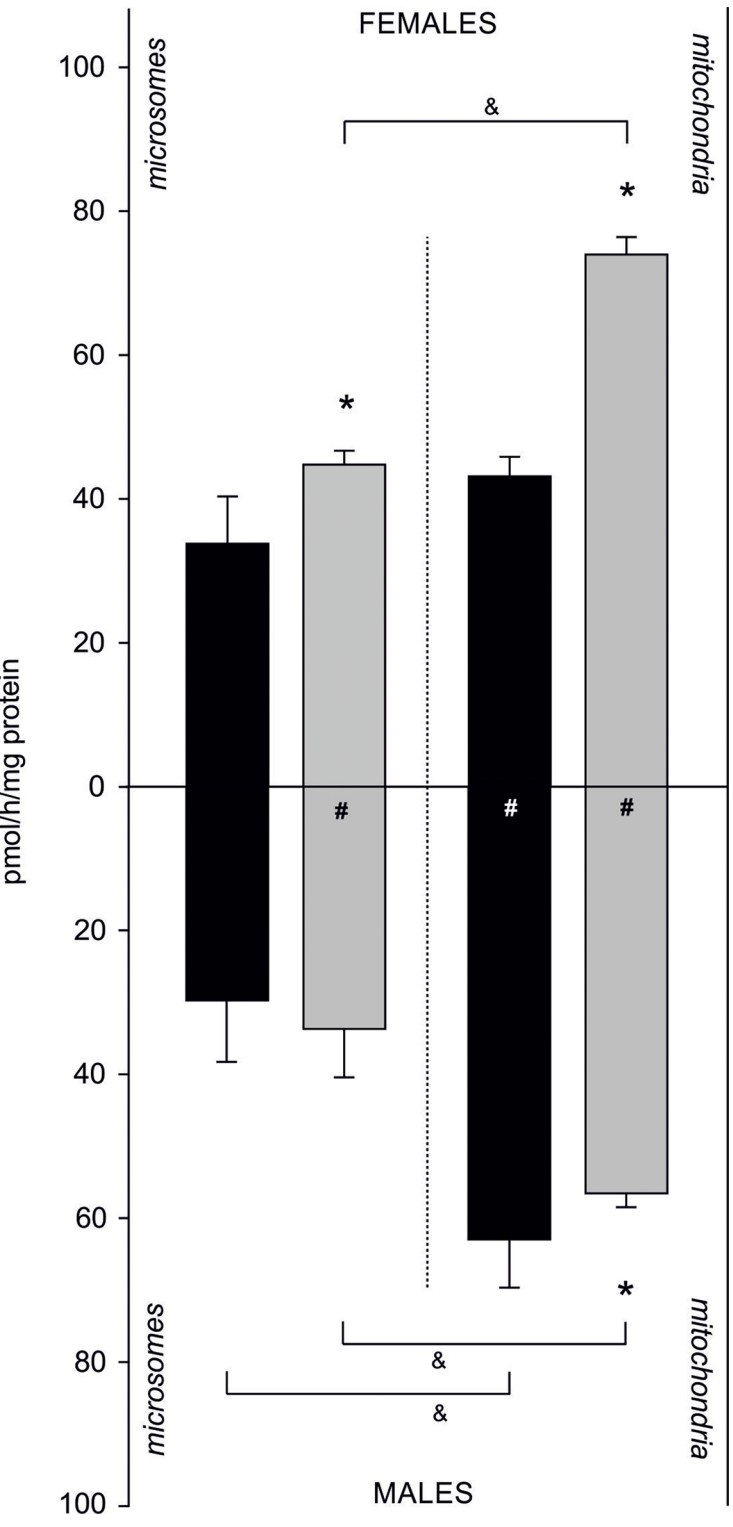

**Figure 4 Sex-related aromatization efficiency in microsomes and mitochondria isolated from gills (black column) and gonads (gray column) of *M. trossulus*.** Data presented as mean ± SD ($n = 3$); $^*p < 0.05$ compared "gill" and "gonads" groups (Kruskal–Wallis ANOVA), $^{\#}p < 0.05$ compared "female" and "males" groups (Kruskal–Wallis ANOVA) and $^{\&}p < 0.05$ compared "microsomes" and "mitochondria" groups (Kruskal–Wallis ANOVA).

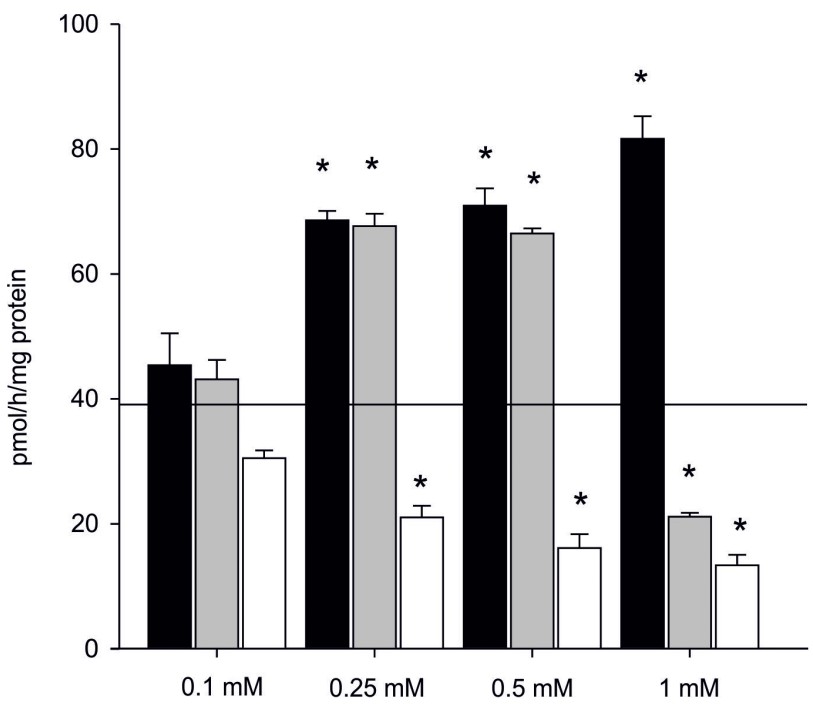

**Figure 5 Effect of inhibitors on aromatization efficiency in microsomes isolated from gonads of *M. trossulus*.** The effect of letrozole is presented in black column, anastrozole in gray column and ketoconazole in white column. Data presented AE in relation to control (black line marks: AE without inhibitors = 39.04 ± 7.34 pmol/h/mg of protein). Data presented as mean ± SD ($n = 3$). Significant differences indicated by *$p < 0.05$ (Kruskal–Wallis ANOVA).

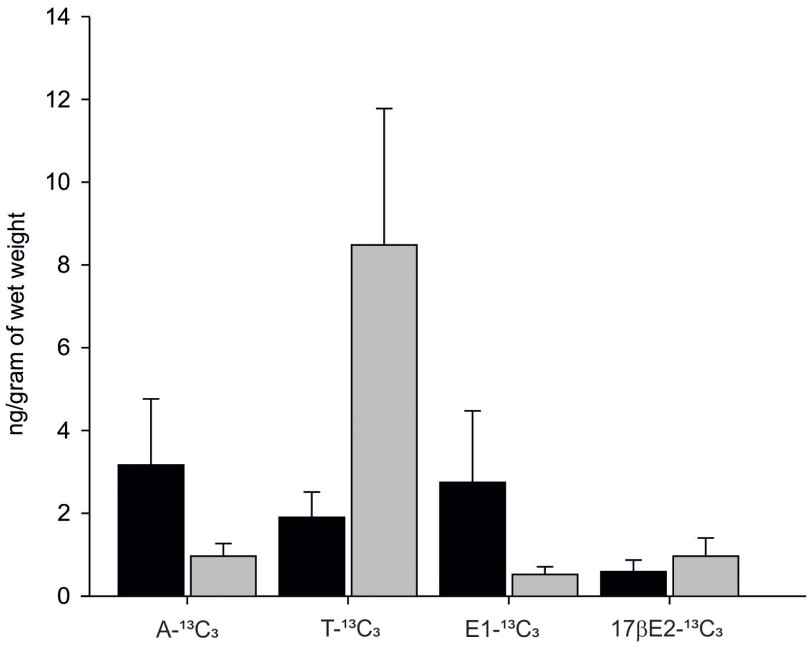

**Figure 6 Levels of A-$^{13}$C$_3$, 4-Androstene-3,17-dione-2,3,4-$^{13}$C$_3$; T-$^{13}$C$_3$, Testosterone-2,3,4-$^{13}$C$_3$; E1-$^{13}$C$_3$.** E1-$^{13}$C$_3$, Estrone-2,3,4-$^{13}$C$_3$ and 17βE2-$^{13}$C$_3$, 17β-estradiol-2,3,4-$^{13}$C$_3$ in *M. trossulus* exposed to 4-Androstene-3,17-dione-2,3,4-$^{13}$C$_3$ (black column) and Testosterone-2,3,4-$^{13}$C$_3$ (gray column). Data presented as mean ± SD ($n = 10$).

using an isotopic method equalled to 0.3–0.8 pmol/h/mg protein (*Barbaglio et al., 2007*). In *Lavado, Janer & Porte (2006)*, an isotopic method with 1β-$^3$H androstenedione as a substrate for detection of aromatase activity in microsomal fraction from gonads and digestive gland of *M. edulis* was applied. At the time of measurement, a possibility of unspecific cross-reactivity was not taken into consideration, hence the authors presented their results as a measurement of steroidogenic P-450-aromatase activity. In spite of methodological similarities, described in *Lavado, Janer & Porte (2006)* enzymatic activities were much lower than presented in our work and ranged from 0.3 to 3.0 pmol/h/mg protein depending on type of tissue and exposure type. The referred measurement of aromatase activity was performed under laboratory controlled conditions at 25 °C. Since comparable low activities were obtained in our study in similar temperature regime, we believe that the main reason behind differences in aromatization rate obtained in our and in *Lavado, Janer & Porte (2006)* is temperature in which the actual measurement was performed. After analysing AE in various temperature regimes in in vitro and seasonally, it became clear that aromatization in bivalves is a temperature-dependent process and temperature is a strong limiting factor for it to occur efficiently. That is especially important when taking into account mussels inhabiting colder temperate areas such as the Baltic Sea. It has to be, however, noted, that bivalves from warmer areas may have different temperature optimum for aromatization to be efficient than those inhabiting colder regions.

In *M. edulis* complex inhabiting temperate and polar waters also gametogenesis belongs to temperature-related processes which is initiated in late winter and proceeds until early summer when spawning takes place. In the Baltic Sea, water usually reaches temperatures of 8–9 °C in spring, thus allowing for various metabolic processes including androgen aromatization to start. During that time the need for estrogens may also increase. In our previous studies performed on blue mussels collected from the Gulf of Gdańsk (Poland), the lowest steroids content was recorded in winter when water temperature is the lowest. In spring, an increase in the amount of estrogen was identified with the highest estrogens concentration found in mussels collected during summer season (*Smolarz et al., 2018*). The estrogen content in tissues decreased in autumn together with a temperature decrease. The highest level of estrogen in mussel tissues found in summer is most likely related not to an increased efficiency of steroidogenesis, but rather to efficient uptake of steroids from the ambient environment as it was already proved by various studies including our own (*Schwarz et al., 2017a*, *2017b*, *2018*; *Smolarz et al., 2018*). Active steroidogenesis occurs only in spring when the level of sex steroids in the ambient environment is low. That may also be related to the fact that the administration of hormones from the outside simply inhibits their synthesis, a phenomenon well-known from research on vertebrates. Optimization (consideration of temperature) of the isotopic method designed for quantification of aromatization allowed for obtaining aromatisation rate in studied blue mussels at levels similar to those available for higher organisms (*Emoto & Baird, 1988*). Interestingly, a strong dependency of aromatization efficiency (due to changes in aromatase activity) on temperature, like the one observed in our study, has already been described in various species displaying temperature-dependent sex

determination such as reptiles, including crocodilians, turtles, and lizards (*Crews et al., 1994*). In these species aromatase activity remains universally low with steroidogenesis often beginning very early. Indeed, at the beginning of the thermosensitive period an increase in aromatase activity appears and is temperature-specific with the temperature window depending on species. In marine and freshwater turtles rising temperatures cause an exponential increase of aromatase activity, whereas in lower temperatures aromatase activity remains low. Distinct levels of aromatase activity drive the differentiation of indifferent gonads into sex-specific reproductive apparatus, that, once established, becomes no longer affected by temperature changes (*Manolakou, Lavranos & Angelopoulou, 2006*). Recent studies also highlight a possibility of epigenetic regulation of the sex determination in reptiles mediated by cold-inducible RNA binding protein (*Georges & Holleley, 2018*). The presence of aromatase (or aromatase-like enzyme) in bivalves has not been confirmed so far either on a protein or on a genetic levels but verification of its (or a similar gene catalysing aromatization in this group of organisms) localization could result in the discovery of mechanisms behind sex determination similar as those in reptiles. According to *Castro, Santos & Reis-Henriques (2005)*, the probability of aromatase gene ortholog being present in Mollusca genome is quite high as the presence of a MHC-paralog gene similar to vertebrates was recently confirmed. The recently published study by *Thitiphuree, Nagasawa & Osada (2019)* suggests that aromatisation, if occurring in bivalves, is not based on vertebrates type aromatase, but on other enzyme from cytochrome P450 family.

Since the production of tritiated water may be related to the presence of aromatase as well as to another enzyme belonging to the cytochrome P450 family, we decided to use known vertebrate aromatase inhibitors (letrozole, anastrozole, and KZ) in order to confirm that aromatisation in bivalves is catalysed by aromatase in the model species. Letrozole affects aromatase in a variety of tissues including human placenta ($IC_{50}$ −11 nM) and rat ovarian microsomes ($IC_{50}$ −7 nM), hamster ovarian cells ($IC_{50}$ −20 nM), human breast ($IC_{50}$ 0.14–0.8 nM) or JEG-3 human *choriocarcinoma* cells ($IC_{50}$ 0.07–0.45 nM) (*Haynes et al., 2003*). It binds to the iron in heme moiety of CYP-450, whereas, the cyanobenzyl moiety partially mimics the steroid backbone of the enzyme's natural substrate androstenedione (*Bhatnagar, 2007*). In non-cellular systems, letrozole is two to five times more potent than anastrozole. Anastrozole hinders human aromatase by 50% at a concentration of 0.043 pg/mL (15 nM) (*Hortobagyi & Buzdar, 1998*). In our study, both letrozole and anastrozole proved to be characterized by low or non-existing inhibitory affinity to the enzyme catalyzing aromatization in bivalves. Atypical mechanism of action of letrozole was also found since the drug not only did not inhibit the reaction, but stimulated aromatization process in tested bivalves. Similarly to letrozole, the lower doses of anastrozole used in our study induced the aromatization rate in mussels, but higher concentration of anastrozole indeed inhibited the process by over 50%. KZ is characterized by direct in vitro inhibition of the human ovarian enzymes 3β-hydroxysteroid dehydrogenase and 17-hydroxylase activity. These two enzymes are proven to be essential for the formation of C-19 steroids: androstenedione and testosterone (*DiMattina et al., 1988*). Furthermore, KZ has been shown to inhibit cholesterol side chain cleavage enzyme

in both the adrenal and testis. *Santen et al. (1983)* concluded that KZ inhibits several other cytochrome P-450-dependent steroid hydroxylases. In vitro studies (*Loose et al., 1983*; *Kunio et al., 1986*) provided evidence that ketokonazole inhibits hydroxylation of deoxycorticosterone and renal 24-hydroxylase suggesting that ketokonazole indeed belongs to nonspecific inhibitors of many cytochrome P-450 enzymes. Ketokonazole is also widely used as antifungal drug exerting its antifungal effect by inhibition of ergosterol biosynthesis and it appears to be a potent inhibitor of triglycerides and phospholipids synthesis in fungi (*Van Tyle, 1984*). In our study, only ketokonazole application brought expected results as it successfully inhibited tritiated water production in mussel gonads in all tested concentrations. Summarizing, selected specific (low spectrum) inhibitors of mammalian aromatase were not effective as inhibitors (letrozole) or only effective when applied in high concentrations (anastrozole), suggesting that the structure of the protein catalyzing aromatization in mussels differs from the one known as mammalian aromatase. Only ketokonazole inhibitory potency was confirmed, but since the pharmaceutical is characterized by a high-spectrum of action it is possible that it can also inhibit aromatase-like protein belonging to the CYP 450 family that happens to catalyse aromatization in mussels. Hence, we believe that the presence of non-specific cross-reactivity with aromatization catalyzed not by aromatase typical for vertebrates but by an enzyme similar in function but different in structure (belonging to CYT P450 family) is occurring in molluscs. The results of recently published molecular identification of steroidogenesis-related genes in scallops suggest that the occurrence of vertebrate type aromatase in bivalves is unlikely (*Thitiphuree, Nagasawa & Osada, 2019*), what is also supported by the results of our study. However, aromatisation (conversion of androgens to estrogens) may still take place via other enzymes from cytochrome P450 family, whether it is CYP19-like ("aromatase-like" as suggested by us) or a protein coded by CYP 3 gene (as suggested by *Thitiphuree, Nagasawa & Osada, 2019*). Thus, more research has to be done in order to identify and describe the protein catalysing aromatisation.

Our results also indicate that in mitochondrial fraction aromatization process appeared to be more efficient than in microsomal fraction. Mitochondrial fraction is protein-rich and contains enzymes more efficiently hydroxylating labeled substrate than those found in the microsomal fraction (*Felty & Roy, 2005*). In vertebrates, steroid synthesis starts from synthesis of pregnenolone from cholesterole, and the reaction is operated by mitochondrial complex of cholesterol desmolase (protein type group CYT P450 - CYP11A) (*Milczarek et al., 2008*; *Miller, 2011*; *Ramalho-Santos & Amaral, 2013*). Synthesized pregnenolone is further relocated to endoplasmic reticulum, where the synthesis of progesterone takes place. Progesterone, depending on the tissue it originates from, then becomes the precursor of various steroid hormones. Based on the final stage of estrogen synthesis, in vertebrates aromatase is active mainly in the microsomal fraction (*Carreau et al., 2003*), but can also be found in the mitochondria of human placenta (*Smith et al., 1999*; *Manolakou, Lavranos & Angelopoulou, 2006*). Our results suggest that aromatase-like enzyme catalyzing aromatisation in bivalves is characterized by similar to vertebrate' aromatase mechanism of action and is in affinity with both the mitochondrial and microsomal fractions regardless the tissue studied. Our results also highlight that

efficiency of aromatisation in female mussels is relatively high in mitochondrial fraction isolated from the gonadal tissue.

Latest studies report that bivalves from genus *Mytilidae* are able to uptake large amounts of progesterone (*Schwarz et al., 2018*), testosterone (*Schwarz et al., 2017b*), and 17β-estradiol (*Schwarz et al., 2017a*) from the ambient environment. This high steroids uptake potential was also confirmed in our study with A-$^{13}$C$_3$ and T-$^{13}$C$_3$ detected in mussel tissues after exposure to labeled substrate. Those androgens were further metabolized to estrogens: estrone-2,3,4-$^{13}$C$_3$ (E1-$^{13}$C$_3$) and 17β-estradiol-2,3,4-$^{13}$C$_3$. According to our knowledge, this is the first comprehensive study fully documenting the occurrence of aromatization process taking place in bivalve tissues. In our previous studies (*Hallmann et al., 2016*; *Smolarz et al., 2018*) testosterone level in *M. trossulus* ranged from 3 to 14 ng/g of wet weight, depending on the season and tissue type. These values correspond well with the levels of T-$^{13}$C$_3$ intercepted by mussels in the in vivo experiment. The amount of naturally occurring estrone ranged from 0.5 to 3.5 ng/g w.w., and also corresponded with E1-$^{13}$C$_3$ concentration in the mussel tissues. The level of natural 17β-estradiol oscillated between one and nine ng/g w.w., but the level of synthesized 17βE2-$^{13}$C$_3$ was much lower, near one ng per gram of wet tissue. At the same time, no sex-related differences in remaining androgens-$^{13}$C$_3$ and synthesized estrogens-$^{13}$C$_3$ were found. As described above, incubation of blue mussels with T-$^{13}$C$_3$ or A-$_{13}$C$^3$ resulted in the formation of T-$^{13}$C$_3$, A-$^{13}$C$_3$, E1-$^{13}$C$_3$ and 17βE2-$^{13}$C$_3$. Such interconversion of steroids is possible when 17β-hydroxysteroid dehydrogenase (17βHSD) is present in mussel tissues and active in the presence of both, androgen and estrogen substrates. In mollusc, 17βHSD catalysed conversion of the estrone to 17βE2 and the enzymatic pathways is already described (*Baker, 2001*; *Scott, 2012*). In the performed exposure experiment, a substrate-specific preference was observed since the uptake of T was three times higher than A. The two hormones can therefore be used as substrates in aromatization and estrogen synthesis. Higher affinity for testosteron observed in blue mussels can possibly be related to the usage of T in a synthesis of 17β-estradiol while androstenedione can be used in estrone (E1) synthesis, the latter occurring in lower concentration in bivalve tissues than E1 (*Smolarz et al., 2018*). In females, estrogens seem to be important in reproduction and immune response regulation (*Stefano et al., 2003*; *Ketata et al., 2008*; *Cubero-Leon et al., 2010*). In male mussels, the role of estrogens may be similar to their role in vertebrates, where E2 initiates testicular maturation and spermatogenesis advancing the gamete' development (*Carreau et al., 2003*).

We have proven that bivalves do have a veritable factory necessary for estrogen production and thus are able to synthesise estrogens, something that has evaded other researchers over the past 70 years. Our results also indicate that bivalves have a great potential of uptaking steroids from the ambient environment in high doses, but only a small part of them will be aromatised. Taking into account mussels ability to concentrate the steroids from ambient environment and their high clearance rate (estimated at 40 mL per hour per animal for T (*Schwarz et al., 2017b*)), all added androgens could have been absorbed and subsequently modified by the mussels. Hence, obtained in experimental conditions conversion rate of androgens to estrogens can be regarded as low, thus

esterification of remaining sex steroids is highly expected. Various authors reported bioaccumulation of 17β-estradiol in the form of fatty acid esters in bivalve tissues (*Puinean & Rotchell, 2006*; *Peck et al., 2007*; *Scott, 2018*). Moreover, in *Schwarz et al. (2017b)*, apart from esters binding T, dihydrotestosterone, 5α-androstan-3β,17β-diol, and 5α-androstan-3α,17β-diol were found to be formed after exposure of mussels to similar doses of labeled T. The failure in detection of estrogen production by *Schwarz et al. (2017b)* can thus be related to the fact that, apart from temperature and season, labeled estrogens peaks (low-yield fraction) could have been missed due to the presence of mentioned above high-yield androgen metabolites possibly reducing chromatogram' readability.

In bivalves estrogens bind to specific receptors but molecular pathway of steroids in general is not yet fully recognized. The presence of ER and estrogen response element (ERE) was indeed confirmed in the genome of the Sydney rock oyster *Saccostrea glomerata* (*Tran et al., 2016*) whereas in the Pacific oyster *C. gigas* cgER ER was found to be homologous in nearly 90% with the human ERα and ERβ located in the nuclei of follicle cells (*Matsumoto et al., 2007*). In *M. edulis* not only the presence of ER2 receptor homologous with human ERβ was reported, but also its activation by estrogens in the early stage of gametogenesis was confirmed (*Ciocan et al., 2010*). The occurrence of different class of nuclear receptors able to bind steroids in mollusks has also been suggested after *Di Cosmo et al. (1998)* and *Di Cosmo, Di Cristo & Paolucci (2002)* proved that the binding proteins for progesterone and 17β-estradiol in *Octopus vulgaris* have the ability to bind to DNA. Binding of vertebrate type estradiol-17β to invertebrate ERs is, however, a low affinity process (*Thornton, Need & Crews, 2003*). Hence, without genomic description of the binding proteins for steroids (*Thornton, Need & Crews, 2003*; *Bertrand et al., 2004*; *Bridgham et al., 2014*) it is not possible to confirm whether estrogens are successfully binding to them and therefore give a biological response in mollusks (*Björnström & Sjöberg, 2005*; *Keay, Bridgham & Thornton, 2006*; *Bannister et al., 2013*; *Schwarz et al., 2017a*). In vertebrates, estrogens may also react in a non-genomic way by attaching to a receptor located on the cellular membrane. The non-genomic effect of estrogens may affect the signaling pathways of MAPK kinases, tyrosine and lipid kinases (*Simoncini & Genazzani, 2003*; *Björnström & Sjöberg, 2005*) and this interaction could be similar in bivalve mollusks. In our study, more efficient aromatization of estrogens in mitochondria than in microsomes isolated from bivalves was found. Vertebrate models proved that estrogens easily penetrate the mitochondria through diffusion and endocytosis and regulate the transcription of the mitochondrial genome in ERE sites (*Felty & Roy, 2005*). In bivalve models ERE regions have not been found so far. However, in DUI species such as *M. trossulus*, estrogens can be involved in the regulation of sex determination linked to mtDNA inheritance pattern associated with sex-ratio bias (*Zouros et al., 1994*). This may explain estrogens synthesis during gonadal maturation and higher aromatization efficiency in spring.

## CONCLUSIONS

In our study, aromatisation of androgens by mitochondria and microsomes isolated from gills and gonads of *M. trossulus* with temperature strongly influencing the efficiency of the

process was detected. High uptake of Testosterone-2,3,4-$^{13}C_3$ and 4-Androstene-3,17-dione-2,3,4-$^{13}C_3$ and their conversion to 17β-estradiol-2,3,4-$^{13}C_3$ and Estrone-2,3,4-$^{13}C_3$ was also identified. Thus, steroids in bivalves can be of endogenic and exogenic origin with estrogen' biosynthesis taking place only in the narrow temperature window (Spring) in mussels inhabiting colder temperate area. The vertebrate aromatase inhibitors such as letrozole and anastrozole had an unusual effect on mussels' AE, with the process being more rapid after their usage. Only the highest dose of anastrozole and KZ inhibited the aromatization process, suggesting that the aromatase-like enzymatic complex belongs to large CYT450 family. This, and the fact that aromatisation was more efficient in mitochondrial and microsomal fractions, mean that the aromatase-like enzymatic complex involved in aromatisation in bivalves is characterized by a different structure than vertebrates aromatase.

### Funding
This work was funded by the grant from the Polish Ministry of Science and Higher Education N N304 07440, DS of the Department of Marine Ecosystem Functioning, University of Gdańsk, Poland (prof. Maciej Wolowicz) and prof. Ryszard Smoleński (Head of the Department of Biochemistry, Medical University of Gdańsk, Poland). The funders had no role in study design, data collection and analysis, decision to publish, or preparation of the manuscript.

### Grant Disclosures
The following grant information was disclosed by the authors:
Polish Ministry of Science and Higher Education N N304 07440.
DS of the Department of Marine Ecosystem Functioning, University of Gdańsk, Poland.
Head of the Department of Biochemistry, Medical University of Gdańsk, Poland.

### Competing Interests
The authors declare that they have no competing interests.

### Author Contributions
- Anna Hallmann conceived and designed the experiments, performed the experiments, analyzed the data, contributed reagents/materials/analysis tools, prepared figures and/or tables, authored or reviewed drafts of the paper, approved the final draft.
- Lucyna Konieczna analyzed the data, contributed reagents/materials/analysis tools, approved the final draft.
- Justyna Swiezak prepared figures and/or tables, approved the final draft.
- Ryszard Milczarek approved the final draft and consulted the work with labeled chemicals.
- Katarzyna Smolarz conceived and designed the experiments, performed the experiments, analyzed the data, contributed reagents/materials/analysis tools, authored or reviewed drafts of the paper, approved the final draft.

## Data Availability

The raw data are available in the Supplemental Files.

## Supplemental Information

Supplemental information for this article can be found online at http://dx.doi.org/10.7717/peerj.6953#supplemental-information.

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
