# Peer review of "Aromatisation of steroids in the bivalve Mytilus trossulus"

_PeerJ, doi:10.7717/peerj.6953_

## Round 0.1 · original submission · Major Revisions

While both reviewers agree that your paper has merit and is well written, the methodology as well as some of your assumptions need to be better explained. Please carefully address their suggestions and comments.

Reviewer 1 ·

Basic reporting

The paper by Hallmann et al. is well written and they reported a potential steroidogenesis, particularly aromatization of androgens for estrogen synthesis, in Mytilus trossulus based on the precise experiment using radio-labelled steroids. The authors suggested the occurrence of aromatase activity influenced with temperature, pH, gender, maturity in this species, and the specific conversion of androgen to estrogen.

Experimental design

The experiments were well designed and the results from these experiments provided an accurate and convincing data.

Validity of the findings

The findings are significantly valuable to discuss about endogenous estrogen synthesis in bivalve mollusks because invertebrate steroidogenesis is still highly controversial.

Additional comments

L553-555: The authors argued that bivalves are able to synthesize vertebrate estrogens. This argument is uncertain and supposed to mislead readers to synthesis of estrogens, which is completely identical to vertebrate estrogens, in bivalves. The inhibitory effects of inhibitors to aromatase on aromatization reaction in microsomes of gonad were not observed as the authors expected and vertebrate type estradiol-17ß have been reported to bind to invertebrate estrogen receptors with low affinity, which is discussed in binding sites of ER to E2 in detail by Thornton group. These facts disagree with endogenous synthesis of vertebrate type estrogens in bivalves although the possibility of synthesis of estrogen-like compounds cannot be ruled out in bivalves. Recently Thitiphuree et al. (J Steroid Biochem Mol Biol, 186, 22-33, 2019) proposed potential steroidogenic pathway including aromatization.

·

Basic reporting

The manuscript reports results from in vitro measurements of mussel activity using radiolabelled steroids. It aims at demonstrating the activity of aromatisation in the mussel Mytilus trossulus by measuring the release of radiolabelled 3H2O from 1β-3H androstenedione and by the identification of estradiol or estrone in mussel tissues resulting from the metabolisation of aromatisable substrates.
This is a very interesting subject worth investigating.
The manuscript is well presented and written and conforms to essential requirements.
However, there are some methodological weaknesses and the existence of interpretations that are not fully supported by the facts. The manuscript must therefore be corrected before publication.

Experimental design

The M&M part need more precise description. Please provide explanations to the following questions.
l. 199-200: How many mitochondria and microsomes fractions were prepared per sampling time (i.e. how many 25mL of tissues were extracted) ?
l.207 and l. 222: It is mentioned that these fractions were fractionated into 1mg portions. 1 mg of what? How was it measured?
l. 180. According to M&M “In vitro analyses were performed on organisms collected in the period from 2012 to 2018.” However, Fig 3 show that there were large variations of activities between sampling times (seasonal effects). How did you ensure that the microsomal and mitochondrial fractions used (each 1mg extracts) were comparable between assays?
Were the results from n=5 shown in fig 3 and 4 from different extractions (different fractions)?
Enzymatic activity: Please show the results of the controls (1) with denatured (heated) microsomal fraction and (2) no NADPH. This is very important as no inhibitors are identified.
Please show results with consistency using a single unit (pmol/h/mg protein) in all figures.
Fig 6: What were the sex of the sample used?
Fig 6: How many mussels from the two experiments (15 unsexed or 10 sexed mussels) were used in the showed results? How come that n=10? Are those results from only the second experiment? Please specify.
The symbols are misplaced in the figure 6. Please use the same form as the figures you published in 2018.
There is no indication of any difficulties to determine the sex of the sample mussels. However, some were sampled in November where usually no sex can be easily determined especially with the method used in this work. Please explain.
l.267: Steroids-13C3 identification: please provide detection/quantification limits.
Please explain the sentence (l.526-527): “Unfortunately, due to the methodological restrictions simultaneous detection of androgens and estrogens after exposure to labelled substrates was not possible.”

Validity of the findings

A number of questions and remarks may arise considering the discussion and conclusion.
As noted by the authors (l.105-108), “non-specific cross-reactivity cannot be excluded since the reaction can be catalysed by aromatase as well as by any random enzyme belonging to the group of the cytochrome P450 family that contributes to oxygenations and loss of the methyl group at C-19 position.” This means that it is unsure that any aromatisation of steroids occurred. It would have been necessary to identified the products. Thus what the authors called aromatase activity should be designated with more caution.
Validity of the findings are subjected to methodological weakness as specified above. Full description of the methods is required in order to better judge the strength of the results.
l.413-430. Temperature effect: According to your previous published paper (2018, table 1), the temperature of your samples was close to optimal in February 2013 at your sampling location (7,8°C) and too high during the other months (14,5-17,5°C). Please discuss the implication of these findings considering that you reported low levels of steroids (and especially E2) in February 2013 (winter time) and high in May 2012 (fig 3 and previous paper). Your discussion is not in accordance with this optimum in winter.
l.455-478: Deduction from lack of inhibition: This (very) long paragraph discuss the fact that there were no inhibitory actions of several of known vertebrate aromatase inhibitors on the measurements of the potential mussel aromatase activity (release of radioactive water) in the present study. From the absence of the vertebrate aromatase in the mussel extract, the authors conclude that (what may or not be) the mussel aromatase is different. It is very surprising to conclude in the existence of a particular protein by demonstrating the absence of another! (It is also a bit strange to write in a scientific journal that “we believe” as a concluding remark. The evidence is missing).
l.518-552: Are the mussel able to aromatise steroids in vivo? This paragraphs discuss the results from the exposure to A-13C3 and T-13C3. Please explain why the chromatograms could not allow the identification of the radiolabelled compounds simultaneously.
As noted (l.562-564), mussel can very rapidly metabolise steroids into esters. How come that none was observed here? Why did not you include any saponification step in your method?

Additional comments

The present description of the method in this study and the results does not allow to conclude on the existence of any aromatase in the mussel. Further information and description are needed.
The authors might provide satisfactory answers. If not, conclusions should be changed accordingly.

---

## Round 0.2 · accepted · Accept

Congratulations, your revisions to the manuscript have satisfied both reviewers, they and I have found your revised description of the experimental design and interpretation of results to be acceptable for publication.

# Reviewer 1 ·

Basic reporting

The authors considerably addressed the issues on synthesis of vertebrate type estrogen in bivalve species in Discussion.

Experimental design

Basic information on the specimen and sample preparation was added.

Validity of the findings

The authors revised their argument on endogenous vertebrate type-estrogen synthesis according to the review comments, which can be understood.

Additional comments

The author's response to the review comments and revision satisfied me. I would recommend the manuscript to be published.

·

Basic reporting

The answers provided are generally satisfactory (although some seem to leave some uncertainties). The article seems to me to be publishable as it stands.

Experimental design

the complementary data provided after the corrections are satisfactory.

Validity of the findings

The authors' interpretation is valid.

Additional comments

Thank you for your efforts to answer all the questions. And congratulations on your work.